# Patellar Tracking in Total Knee Arthroplasty—Influence on Clinical and Functional Outcome

**DOI:** 10.3390/diagnostics12051082

**Published:** 2022-04-26

**Authors:** Sebastian Dahlmann, Katharina Ziegeler, Anett Mau-Möller, Wolfram Mittelmeier, Philipp Bergschmidt

**Affiliations:** 1Department of Radiology, Charité Universitätsmedizin Berlin, Luisenstraße 7, 10117 Berlin, Germany; katharina.ziegeler@charite.de; 2Orthopaedic Clinic and Outpatient Department, University Medicine Rostock, Doberaner Straße 142, 18057 Rostock, Germany; anett.mau-moeller@uni-rostock.de (A.M.-M.); wolfram.mittelmeier@uni-rostock.de (W.M.); philipp.bergschmidt@kliniksued-rostock.de (P.B.); 3Department for Orthopaedic Surgery, Trauma Surgery and Hand Surgery, Südstadt Hospital Rostock, Südring 81, 18059 Rostock, Germany

**Keywords:** total knee arthroplasty, anterior knee pain, patellar tracking

## Abstract

Anterior knee pain is a common problem after primary total knee arthroplasty (TKA). The aim of this study was to find parameters in patellar positioning which influence the clinical and functional outcome after TKA. Included were 59 patients who underwent TKA, of which three patients were treated bilaterally (*n* = 62 included knees). In a periodical follow-up of up to 5 years, each patient had to answer three questionnaires (HSS, WOMAC, SF-36) and underwent three radiographies of the knee (including merchant view) and a clinical examination, including Range Of Motion (ROM). All radiographs were evaluated by a single observer blinded to clinical data, who collected multiple parameters of sagittal and axial patellar alignment including newly developed methods for measuring patellar shift and tilt. Depending on the measurement results, three groups were built for each parameter and the influence on the outcome was determined. A lateral patellar tilt of more than 4° resulted in lower scores for both the HSS and WOMAC. The rarely investigated patellar facet angle showed a significantly inferior clinical and functional outcome in late follow-up of >24 months if lower than 142°, possibly due to progressive osteosclerotic changes of the patella caused by increased contact stress with corresponding patellar morphology. No significant difference was found for all other parameters. The newly developed method for measuring patellar shift has proven to be a valuable and easy instrument in the postoperative setting.

## 1. Introduction

Osteoarthritis of the knee is a common disease with a high socio-economic impact. The estimated prevalence in Germany is 23.8% (2012) [1]. Total knee arthroplasty (TKA) has become a safe and common treatment [2]. Attempts to treat osteoarthritis of the knee joint with surgical techniques date back to the 18th century [3], but prototypes of modern prosthesis systems were not developed until the middle of the 20th century [4,5]. Subsequently, continuous progress in prosthesis design resulted in optimized implant fixation and an improved clinical outcome, including the use of new materials with a longer durability. Nevertheless, there are still symptomatic patients today. One of the main symptoms is anterior knee pain. Although there has been much investigation regarding the risks and determinants associated with anterior knee pain after total knee arthroplasty, no single factor could be specified as it most likely is a multifactorial problem [6,7]. The main subject in this discussion is the femoropatellar joint, which was underappreciated in the early days of TKA [8]. Patellar tracking became more prominent and patellofemoral alignment is regarded to be more important in TKA today [9]. The causes of patellofemoral pain can be divided into functional causes, such as muscle imbalances, and mechanical causes, e.g., rotational errors or overstuffing of prosthesis components, resulting in patellofemoral instability and maltracking [10,11]. Although the influence of malpositioning of the patella has been increasingly investigated since the 1990s, there is still no consensus on its importance for the clinical outcome [9,12,13,14,15].

The aim of this retrospective study was to use X-ray-based indices which determine patella tracking and to examine their influence on clinical und functional outcome.

## 2. Materials and Methods

Based on the approval of the local ethics committee, this study comprised 59 patients who underwent TKA. Three patients were treated bilaterally. A total of 62 TKA could be included in this study. Inclusion criteria were total knee arthroplasty in case of osteoarthritis of primary and secondary etiology. Exclusion criteria were infections, osteoporosis, manifest tumor diseases or deformities and instability that did not allow the use of the available prosthesis systems.

### 2.1. Prosthesis Systems

Three different TKA systems were used. All of them were implanted with cement (Refobacin^®^ Bone Cement, Zimmer Biomet Deutschland GmbH, Freiburg i. Breisgau, Germany). The Genia^®^ system (ESKA GmbH, Lübeck, Germany) with an ultracongruent insert is also suitable in case of partial or total loss of the posterior cruciate ligament. Both the femoral and tibial components are made of a cobalt-chrome-molybdenum alloy. The Multigen Plus System (Lima Deutschland GmbH, Hamburg, Germany) is a posterior cruciate ligament preserving system and was implanted either in a metallic version with a femoral component made of cobalt-chromium-molybdenum alloy or in a ceramic version with a femoral component made of BIOLOX^®^ Delta ceramic. The tibial component is made of titanium alloy in all cases. Since a comparative study involving the same patient population has already shown that the basic outcome of all three prosthesis systems is comparable and does not show any significant difference, a mixed analysis within the scope of this study appears permissible [16].

### 2.2. Surgical Procedure

All total knee endoprostheses included in this study were implanted by two experienced surgeons who had already gained sufficient experience with the implantation of the Genia^®^ and Multigen Plus knee. After “single-shot” antibiosis (1.5 g cefuroxime i.v.) and application of a tourniquet, a medial parapatellar approach according to Payr was used. To provide a clearer view of the knee cavity and the tibial plateau, the patella is inverted laterally after opening the joint capsule and the knee joint is flexed by 90°. Using the respective implantation instruments, the osseus implant site is now prepared piece by piece. After sufficient soft tissue balancing, removal of osteophytic attachments, smoothing of the lateral facet and denervation of the patella, the range of motion in flexion/extension, joint stability as well as patellofemoral alignment are tested during motion using trial prostheses. A lateral release was not performed routinely. After selection of the prosthesis size the definitive prosthesis is implanted using PMMA cement (Refobacin Plus Bone Cement, Biomet Deutschland GmbH, Berlin, Germany). In the end, the final function check is followed by the insertion of one redon drain, wound closure and radiological control of the implant position using a C-arm.

### 2.3. Radiologic Evaluation

X-rays of the knee joint in anterior-posterior (a.p.), lateral and merchant (45° flexion) view were performed preoperatively and at defined intervals as part of the postoperative check-ups (5 days and 3/12/24/60 months postoperatively). Based on the radiographs, radiological evaluation was performed to quantify patella tracking.

#### 2.3.1. Sagittal Alignment

The images in lateral projection with knee flexion between 30° and 45° (dependent on patient compliance and experience of technical assistant) were used to assess vertical positional deviations of the patella cranially (patella alta) or caudally (patella infera). Three different indices were measured (see Figure 1).

The Insall–Salvati Index, which is largely independent from the joint position, is calculated as the quotient of the distance between the poles of the patella and the length of the ligamentum patellae from the caudal patellar pole to the insertion at the tibial tuberosity [17]. Normal range is 0.8–1.2 [18].

Instead of the pole distance, the Blackburne–Peel Index measures the articulating surface of the patella and relates it to a perpendicular from the lower pole of the patellar articulating surface to the tibial plateau [19]. Normal range is 0.6–1 [18].

The Caton–Deschamps Index is measured in a similar way, but instead of the previously described perpendicular, the distance from the caudal pole of the patellar articulation surface to the ventrocranial boundary edge of the tibia is measured and set in relation to the patellar articulation surface [20]. Normal range is 0.6–1.2 [18].

#### 2.3.2. Patellar Shift

Deviations of the patella in the horizontal axis were measured using the patella tangential radiographs. The measurement of axial linear patellar displacement after TKA can be difficult, mainly due to the loss of the typical bony landmarks required for measuring patellofemoral congruence. In 2013, Metsna published the “Patellar Shift Index” [21], a new method to facilitate the measurement of patellar mediolateral alignment in relation to the trochlear groove after TKA (see Figure 2).

First, two parallels to the trochlea facets (line 1 and line 2) and a connecting line between the apices of the anterior femoral condyles (line 3) are drawn. The distance between the intersection points S1 and S2 corresponds to the trochlear width. Then, the maximum mediolateral extension of the patella is determined (line 4) and the mid-perpendicular of line 3 is constructed (line 5). Line 4 is also halved and a perpendicular is dropped from that point onto line 3 (line 6). The distance between line 5 and line 6 corresponds to the patellar shift (see Picture 12). For the purpose of better comparability and independence from any X-ray magnification effects, the “Patella Shift Index” (PSI = Patella Shift/Trochlear Width) is calculated. Negative signs indicate a medial and positive a lateral shift [21].

In addition to the measurement of the PSI to determinate patellar shift, we developed a new method that allows a shift evaluation independent of overlaps of important landmarks in the axial X-ray image due to the implanted endoprosthesis or modification of these landmarks as a result of joint facet resections or stress-induced remodeling (see Figure 3). In contrast to Metsna, the actual patellar articulating surface is taken into account. For example, a superimposing femoral component can lead to a difficult determination of the patellar ridge.

First, a facet parallel to the lateral and medial trochlea is drawn, based on the “Patellar Displacement” method by Gomes [22] (line 1 and line 2). Afterwards, a connecting line between the apices of the femoral condyles (line 3) is constructed. From the intersection of lines 1 and 2, a line perpendicular to line 3 is built (line 4). The point where line 4 intersects the trochlea groove defines the lowest point of the pit. To ascertain the deepest point of the patellar ridge, tangents to the lateral (line 5) and medial (line 6) retropatellar facets are constructed and their intersection point is determined. From this point, a parallel to line 4 (line 7) is drawn. The distance between line 4 and line 7 represents the patellar shift. Values of lateral deviations are given with a positive sign, medial ones with a negative sign. Deviations between −5 and +5 mm were defined as normal [23].

#### 2.3.3. Patellar Tilt

The patella tilt is believed to be involved in the development of anterior knee pain [9,14]. Grelsamer published a method for standardized measurement of the patellar tilt in 1992 [24]. On patella tangential X-rays, a line connecting the medial and lateral patellar poles is drawn and the angle between this line and a horizontal line is determined. However, this method is not very resistant to varying leg rotations. For this reason, the measurement of the patellar tilt according to Gomes [22] was used. It is largely similar to the method by Grelsamer, but instead of the horizontal it uses a line between the apices of the femoral condyles as femoral reference line (see Figure 4) [22]. Openings of the angle to the lateral are given with a positive sign, openings to the medial with a negative sign. Values up to ± 5° are considered to be normal [9].

Since there are sometimes difficulties in determining the mediolateral patellar axis by means of the poles due to osteophytic attachments and anatomical peculiarities, a new standardized measurement procedure for evaluating the patellar tilt was developed (see Figure 5).

First, the deepest point of the patella ridge is determined by applying two tangents to the retropatellar facets. Based on this, a vertical section is constructed 10 mm medially and laterally (line 1 and line 2), starting at the posterior patellar edge and ending at the anterior border. The centers of both sections are identified (S1/S2) and connected with each other (line 3). Finally, a connecting line is formed between the apices of the femoral condyles (line 4). The patellar tilt corresponds to the angle between line 3 and line 4. This method requires a sufficient identification of the anterior-posterior margins of the patella. Since there is sometimes an overlap by prosthetic components posteriorly, it is not applicable in every case. The cut-off for normal results is also set to ±5°.

#### 2.3.4. Patellar Morphology

To ascertain the patellar shape type and to distinguish patellar euplasia from dysplastic variants, the patellar joint surface angle (facet angle) described by Christiani was measured. This angle is formed by the patella ridge and each the medial and lateral joint facet [25] (see Figure 6). On axial radiographs in 60° knee flexion, this angle should be between 120° and 139°. Values < 115° and ≥ 145° are considered as dysplastic [26].

Osteophytes are one of the basic radiographic signs in osteoarthritis diagnostics [27]. During surgical joint replacement, osteophytes are usually removed. Left lateral osteophytes may lock and lead to compulsory guiding of the patella. This results in patellofemoral constraint, which is a common cause of functional limitations and postoperative pain in the femoropatellar joint [11]. A quadrant model was used to classify the lateral bony attachments (see Figure 7). The horizontal division is made at the level of the apices of the femoral condyles. The vertical dividing line runs through the center of the lateral femoral condyle.

### 2.4. Clinical Evaluation

The patient collective was clinically evaluated preoperatively in a standardized manner and at defined intervals of 5 days and 3/12/24/60 months postoperatively. The measurement point five days after surgery was omitted because clinical improvements during this period may be variable and may be masked by postoperative pain. Evaluation of the checkpoint 60 months postoperatively was omitted because of the large amount of missing clinical score data (e.g., due to disability, death or prosthesis explantation). Functional criteria as well as physical and psychological aspects were assessed. For this reason, scores that focus on disease-specific subjective functional and objective findings were selected. This includes the arthrosis-specific Hospital of Special Surgery Score (HSS Score) and the Western Ontario and McMaster Universities Osteoarthritis Index (WOMAC Index). On the other hand, the Short Form 36 Score (SF-36 Score) was selected as a score which is not specific to a particular disease and assesses the health status and quality of life. The sum scores of HSS, WOMAC and SF-36 are each composed of multiple subscores, which have also been separately analyzed.

In addition, the range of motion (ROM) of the knee joint was evaluated, since sufficient joint kinematics are decisive for the functional outcome after TKA. After TKA, a flexion of at least 100°–120° should be aimed for, as this is necessary for everyday activities, such as climbing stairs and personal hygiene [28,29].

### 2.5. Statistics

The measurements of the previously stipulated indices/values for the evaluation of patellar alignment were carried out on standardized X-ray images of the knee joint or patella. In order to reduce measurement variances, the recorded (index) values were averaged per patient if evaluable images were available for at least three of the five postoperative control points. The reliability of the data was checked by calculating the intra-class correlation according to the SPSS model “two-way mixed single measure”. A good to very good intra-class correlation with coefficients between 0.813 and 0.964 was found across all indices/values.

Subsequently, three patient groups were formed for each index. The intermediate group (group II) included all values in the range of mean ± 0.5 SD. Values above the cut-off were assigned to group III, values below to group I. The groups (determined based on the radiological values) were then allotted to the corresponding clinical score results (including subscales) for 3, 12 and 24 months postoperatively.

Calculation of significant differences across the three groups was performed using the Kruskal–Wallis test or the Mann–Whitney U test to compare the groups in pairs. Due to the large number of pairwise comparisons for each index/value (*n* = 135), the significance level was adjusted using the Benjamini–Hochberg method [30].

## 3. Results

The results of the radiological evaluation after forming three groups for each index/value are shown in Table 1. Since there is no normal distribution of the values, they are given in median and range.

When comparing the clinical score results according to the grouping of the patellar tilt defined by Gomes, there are several significant differences between the patient groups I and II in both the HSS and WOMAC score (including subscores) 24 months postoperatively (*p* ≤ 0.007 and *p* ≤ 0.012, respectively) and occasionally 12 months postoperatively (*p* ≤ 0.012 and *p* ≤ 0.016, respectively), including the Range of Motion (*p* = 0.016). The individual results are listed in Table 2.

With regard to the facet angle, both the arthrosis-specific HSS score and the WOMAC score show significant differences between groups I and II (*p* ≤ 0.011) and I and III (*p* ≤ 0.004) in the late follow-up after 24 months and singularly early postoperatively after 3 months (*p* = 0.001). The results are shown in detail in Table 3.

The sagittal patellar alignment, trochlear width, PSI, newly developed method for determining patella shift and tilt and lateral bony overhang as well as Range Of Motion showed no significant difference between the groups after adjustment of the significance level with the Benjamini–Hochberg method.

## 4. Discussion

Significant differences in the clinical outcome were found for patella tilt, but the results are inhomogeneous. Measurements using the newly developed method and the method according to Gomes must be considered separately. In the former, no difference between the groups is apparent, although the values of group I with a range of −15.3°–−4.35° lie outside the norm lower limit of −5° [9], while the results of groups II and III are within the norm. The situation is different with Gomes’ method [22]. Here, significant differences between groups I and II in the medium-term follow-up after 12 and 24 months can be seen in the WOMAC and HSS scores as well as in the ROM after 12 months. However, this discrepancy in results between these methods may be due to the fact that highly tilted and apparently symptomatic patellae could not be measured with the newly developed method, since the anatomical landmarks for the measurement could not be identified correctly due to superposition. The number of measurable knee endoprostheses was, therefore, *n* = 40 with the newly developed method and *n* = 56 based on Gomes’ measurement. This explains the high deviation of the median values in group I according to the respective method and has to be recognized as limitation. The median in group I was −13.28° with Gomes’ method and −5.34° with the newly developed method. The maximum medial tilt was 5.7° and 5.23°, respectively, and was, thus, only slightly above the normal limit of 5°, which may explain the good clinical outcome in group III.

Previous results on the influence of axial patella malalignment on the clinical outcome are controversial. In a study of 234 patients, no effect of postoperative patellar shift or tilt on pain and ROM was found [12]. This was confirmed by van Houten, who also found no influence of patellar shift and tilt on the anterior knee pain [15]. Another retrospective study including 138 patients demonstrated an association of maltracking with a significantly worse clinical outcome [9].

In the investigated patient collective of this study, there was no significant difference between the patient groups regarding patellar shift, measured with the newly developed method or by calculating the PSI. As a limitation, it must be said that the shift values according to the newly developed method were almost completely within the generally accepted interval of +−5 mm [9] with a range of −5.28–4.67 mm and significant differences may not be expected.

We also found no significant difference in the outcome of groups depending on the sagittal patella position, regardless of the measurement method. This is consistent with previous results from different studies [15,31,32].

The influence of the facet angle on the outcome after TKA has not yet been clearly investigated. Significant group differences in the HSS and WOMAC scores were found in the follow-up after 24 months, both in terms of the cumulative scores and in function and pain subscales. The results in group I were generally lower than in the other two groups. Lower facet angles increased the comparative stress and led to increased osteosclerosis in the medium term. Furthermore, a correlation between patellar morphology and altered femoropatellar contact stress in a study of 157 knee joints without patellar back surface replacement has been described [33]. Similar results and conclusions were seen in a different study including 76 TKAs. In addition to an increase in patellar tilt and an increasing prevalence of anterior knee pain, this study also showed increased patellar osteosclerosis with lower facet angles (<126°) in follow-up > 3 years [34]. In summary, patellar resurfacing may be considered in these cases, as lower facet angles are associated with increased incidence of anterior knee pain in the long-term follow-up.

Lateral osteophytes of the patella are common in the pathogenesis of osteoarthritis [35]. In the present study group, they were reduced intraoperatively in 17 cases as shown by radiography. Nevertheless, all measured radiographs showed osteophytes on the patella, some of which increased in size during the follow-up. The most common location of such lateral osteophytes was in the upper outer quadrant (corresponds to group II), but over time, they tended to be located in the lower outer quadrant (corresponding to group III). It is possible that here in particular, due to mechanical reasons, maltracking could be locked and the outcome could deteriorate [11]. Nevertheless, there was no significant difference between the groups throughout the study.

Finally, some limitations and sources of error must be taken into account when considering the results. With 62 TKAs, the study covers a rather small patient collective. Only in 25 patients (40.3%), X-rays were available for each follow-up appointment, e.g., due to TEP revision in case of infection, need for care or death. In some cases, X-rays were difficult or impossible to evaluate due to poor image quality, suboptimal projection or overlapping of important anatomical landmarks. Non-joint specific scores, such as the SF-36, can be influenced by other comorbidities that have been present from the beginning or newly appeared during follow-up. Finally, the retrospective nature of the study and the radiological evaluation by only one single observer can be taken as possible limitation.

## 5. Conclusions

Overall, two factors were identified that have a significant impact on clinical and functional postoperative outcome. Patellar tilt and the patellar facet angle showed an influence on the outcome, while a deviation in the sagittal plane did not affect the results. Lateral patellar tilt > 4° was shown to lead to poorer results in both the HSS and WOMAC scores, which is consistent with the results of previously published studies [9,14]. The results were even clearer when the facet angle was considered. After 24 months, significant differences in pain, function and total score of HSS and WOMAC score were found with a reduced facet angle. Progressive osteosclerotic changes of the patella due to increased contact stress with corresponding patellar morphology may be useful for this finding [33,34].

The newly developed method for determining patellar shift has proven to be a good instrument for measuring the mediolateral translation of the patella, especially in the postoperative setting with sometimes difficult or impossible to define measurement points. Further investigations on reliability/validity and intra- and interobserver variability are needed.

## Figures and Tables

**Figure 1 diagnostics-12-01082-f001:**
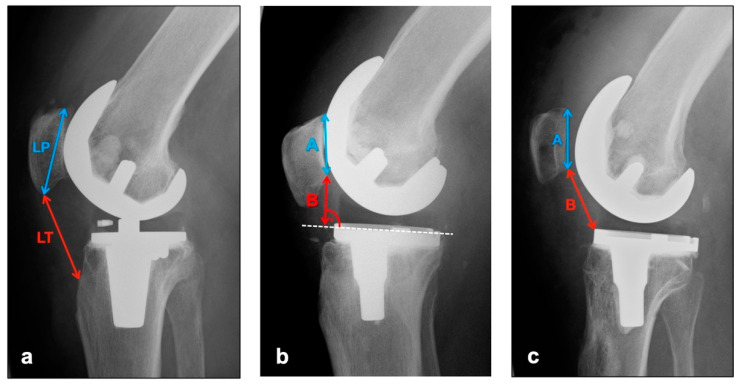
Measurement of patellar height with (**a**) Insall–Salvati Index (Quotient LT/LP), (**b**) Blackburne–Peel Index (Quotient B/A), (**c**) Caton–Deschamps Index (Quotient B/A).

**Figure 2 diagnostics-12-01082-f002:**
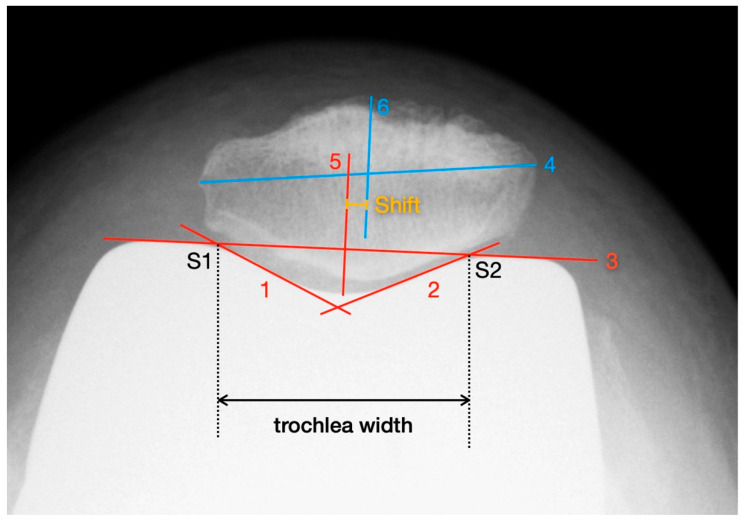
Measurement of patella shift with the method described by Metsna.

**Figure 3 diagnostics-12-01082-f003:**
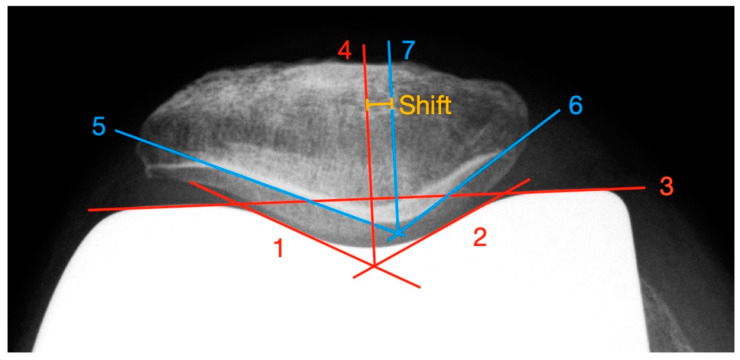
Measurement of the patella shift with the newly developed method.

**Figure 4 diagnostics-12-01082-f004:**
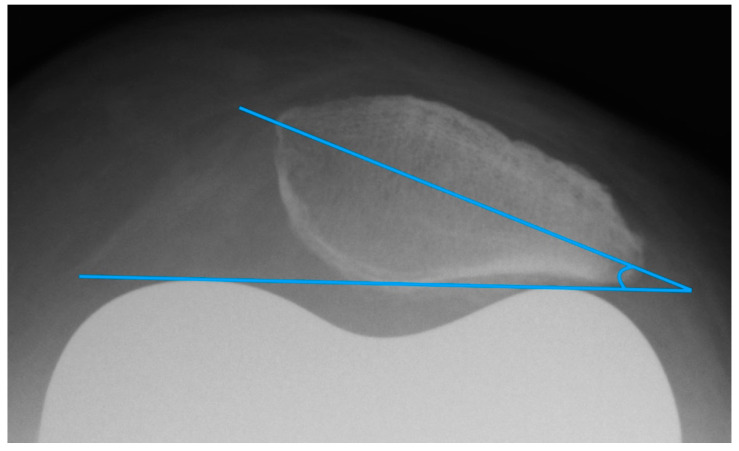
Measurement of the patella tilt with the method of Gomes.

**Figure 5 diagnostics-12-01082-f005:**
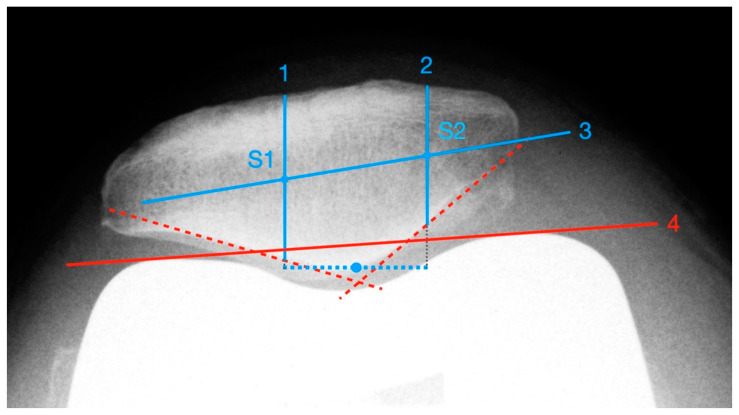
Measurement of patella tilt with the newly developed method.

**Figure 6 diagnostics-12-01082-f006:**
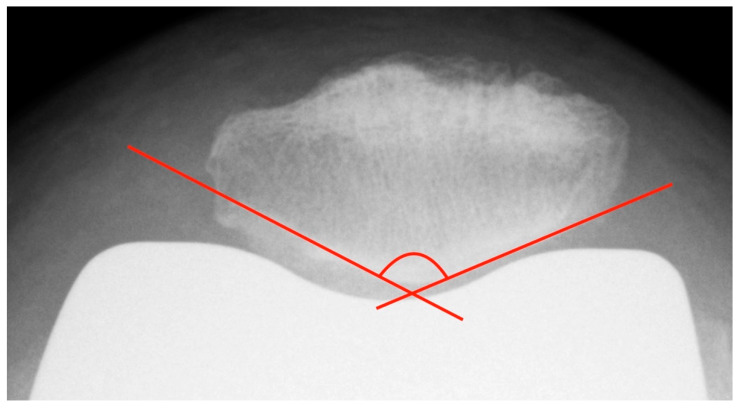
Measurement of the patellar facet angle with the method of Christiani. Additionally, the image shows an example of possible difficulties in determination of patellar facet tangent, in this case due to bony superimposition at the medial facet.

**Figure 7 diagnostics-12-01082-f007:**
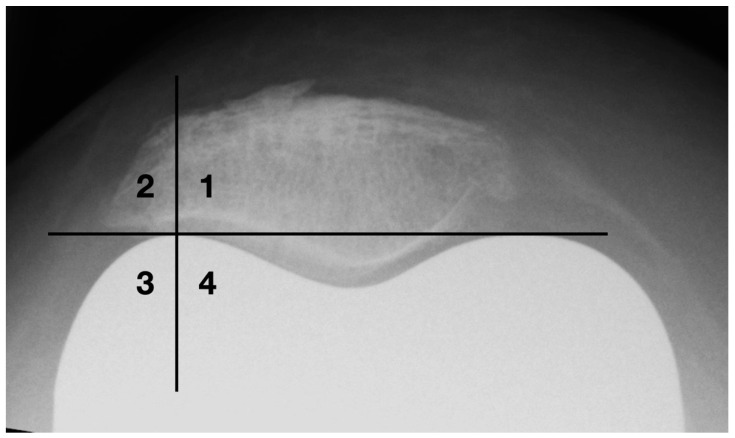
Classification of osteophytes with a four-quadrant-model.

**Table 1 diagnostics-12-01082-t001:** Median, minimum and maximum values of the measured indices/values in groups I–III.

Index/Value	I	II	III
**Insall–Salvati Index**	*n* = 19	*n* = 26	*n* = 15
	0.834 (0.748–0.897)	0.973 (0.910–1.036)	1.146 (1.056–1.570)
**Blackburne–Peel Index**	*n* = 19	*n* = 27	*n* = 14
	0.674 (0.559–0.724)	0.788 (0.733–0.834)	0.898 (0.845–1.07)
**Caton–Deschamps Index**	*n* = 21	*n* = 23	*n* = 16
	0.74 (0.597–0.77)	0.826 (0.774–0.88)	0.963 (0.894–1.165)
**Patella Shift Index**	*n* = 17	*n* = 25	*n* = 15
	0.017 (−0.033–0.037)	0.075 (0.052–0.107)	0.165 (0.11–0.222)
**Patellar shift (newly developed method, mm)**	*n* = 11	*n* = 19	*n* = 13
	−3.15 (−5.275–−2.267)	−0.75 (−1.933–−0.025)	1.1 (0.15–4.667)
**Patellar tilt** **by Gomes (°)**	*n* = 10	*n* = 28	*n* = 17
	−13.28 (−39.57–−9.1)	−4.01 (−7.4–−0.8)	1.28 (−0.65–5.7)
**Patellar tilt (newly developed method, °)**	*n* = 14	*n* = 12	*n* = 13
	−5.335 (−15.3–−4.35)	−1.95 (−3.675–−0.64)	1.567 (−0.025–5.233)
**Facet angle (°)**	*n* = 13	*n* = 15	*n* = 14
	139.25 (131.25–142.0)	144.5 (142.75–146.667)	148.775 (147.8–158.75)
**Lateral osteophytes (quadrant)**	*n* = 8	*n* = 47	*n* = 7
	1	2	3

**Table 2 diagnostics-12-01082-t002:** Significant differences between the groups in the clinical scores and the ROM taking into account the patella tilt according to Gomes. In addition to the significance value *p*, the adjusted significance level after Benjamini–Hochberg correction is given. Bold marked group means better clinical result.

	Score	*t* Post-Op	Compared Groups	*p*	Adjusted Significance Level
**WOMAC**	Score	12 m	I–II	0.016	0.018
	Physical function	12 m	I–II	0.009	0.013
	Score	24 m	I–II	0.004	0.006
	Pain	24 m	I–II	0.012	0.015
	Stiffness	24 m	I–II	0.006	0.007
	Physical function	24 m	I–II	0.003	0.004
**HSS**	Function	12 m	I–II	0.008	0.010
	Range of motion	12 m	I–II	0.012	0.016
	Score	24 m	I–II	0.001	0.001
	Pain	24 m	I–II	0.007	0.009
	Function	24 m	I–II	0.001	0.003
	Deformity	24 m	II–III	0.008	0.012
**ROM**		12 m	I–II	0.016	0.019

**Table 3 diagnostics-12-01082-t003:** Significant differences between the groups in the clinical scores taking into account the facet angle. In addition to the significance value *p*, the adjusted significance level after Benjamini–Hochberg correction is also given. Bold marked group means better clinical result.

	Score	*t* Post-Op	Compared Groups	*p*	Adjusted Significance Level
**WOMAC**	Score	24 m	I–II	0.011	0.012
	Pain	24 m	I–II	0.001	0.001
	Pain	24 m	I–III	0.004	0.007
	Physical function	24 m	I–II	0.009	0.01
**HSS**	Function	3 m	I–III	0.001	0.003
	Score	24 m	I–II	0.007	0.009
	Function	24 m	I–II	0.001	0.004
	Function	24 m	I–III	0.003	0.006

## Data Availability

The data presented in this study are available on request from the corresponding author.

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
