# Peer review of "Patellar Tracking in Total Knee Arthroplasty—Influence on Clinical and Functional Outcome"

_diagnostics, 2022, doi:10.3390/diagnostics12051082_

Round 1

Reviewer 1 Report

The paper is well written. It contains several images to describe how all measures were defined, which is really necessary.

My only suggestion is related to the description of the statistical model. The model is not SPSS, so it is better to write:

"According to the two-way mixed model
for a single response variable using 
the SPSS (version xxx)."

Why two factor? One is the individual
that should be the random factor, 
which one is the other factor?
Is the other factor the group defined
later? It should be clear.

Maybe you could also present the non-significative results.

You should review all the text to replace no significant differences by no significant difference.

When you cite Metsna method (line 116), it is necessary to include a reference.

In the conclusion (Line 343), write which is the outcome, because it should be possible to understand which is the conclusion without reading all the paper.

Author Response

Dear reviewer,

The paper is well written. It contains several images to describe how all measures
were defined, which is really necessary. My only suggestion is related to the
description of the statistical model. The model is not SPSS, so it is better to write:
"According to the two-way mixed model for a single response variable using the
SPSS (version xxx)." Why two factor? One is the individual that should be the
random factor, which one is the other factor? Is the other factor the group defined
later? It should be clear.
We thank the reviewer for this interesting suggestion. “Two-way mixed single
measure” is a SPSS-specific model for calculating the intra-class correlation
coefficient based on the ICC type ICC(3,1) by Shrout & Fleiss [1] and ICC(C,1) by
McGraw & Wong [2]. “Two-way” means not two factors, it means that all cases (in the
study’s specific case all knees) have been evaluated by each observer (in this case
only a single observer). Otherwise, it would have been a “One-way”-model. “Mixed”
means that raters have not been picked by coincidence, otherwise it would have
been defined as “random”. At last individual values for each interval have been used
for calculation and not average values, this is defined as “single measure”; otherwise
it would have been “average measure”. We hope that this explanation solved any
unclarities.
Maybe you could also present the non-significative results.
To maintain a clear presentation of the results, we limited ourselves to presenting
significant results. If the detailed results are of interest to readers, we will be happy to
send them on request. Accordingly, we have also formulated this in the Data
Availability Statement.
You should review all the text to replace no significant differences by no significant
difference
We thank the reviewer for this important consideration. We have changed
corresponding text passages.

When you cite Metsna method (line 116), it is necessary to include a reference.
We apologize for any unclarities. It has been corrected accordingly.
In the conclusion (Line 343), write which is the outcome, because it should be
possible to understand which is the conclusion without reading all the paper.
We apologize if there has been any misunderstanding. Actually, it has been shortly
pointed out, which factors led to poorer results in which clinical and functional scores.
We believe that together with the abstract it is well possible for the reader to
understand the method and results of our study in the specific context. If there are
still any unclarities, we friendly invite the reviewer to explain them in detail.
References
[1] P. E. Shrout and J. L. Fleiss, “Intraclass correlations: uses in assessing rater
reliability.,” Psychol. Bull., vol. 86, no. 2, pp. 420428, Mar. 1979, doi:
10.1037//0033-2909.86.2.420.
[2] K. Mcgraw and S. P. Wong, “Forming Inferences About Some Intraclass
Correlation Coefficients,” Psychol. Methods, vol. 1, pp. 3046, Mar. 1996, doi:
10.1037/1082-989X.1.1.30.

Regards

Reviewer 2 Report

There is a lot of work on a similar topic. This work adds nothing new. Other similar studies assess a larger number of patients.

Introduction too short, no good literature review.

Too short discussion, no good literature review.

Author Response

Dear reviewer,

There is a lot of work on a similar topic. This work adds nothing new. Other similar
studies assess a larger number of patients.
Introduction too short, no good literature review.
Too short discussion, no good literature review.
We thank the reviewer very much for his valued opinion. We saw that the previous
work does not cover the full range of possible radiological measurement methods
using the same patient collective. We have tried to include as many established
scores as possible in the study. In addition, we introduced two new measurement
methods to evaluate axial patella alignment. In particular, the novel determination of
the Patella Shift shows very good applicability also for clinical routine and can be
widely used. As the reviewer correctly notes, many papers have been done in this
area, but not necessarily appropriate to the patient population we studied. We have
attempted to select available papers judiciously and place them in the correct
thematic context. We have made multiple revisions to the manuscript and hope that it
now meets the high standards of the esteemed reviewer.

Regards

Reviewer 3 Report

Dear authors,

please see the attached file.

Author Response

Dear reviewer,

Dear authors, Dear Editor, thank you for giving me the chance to evaluate your work.
This is an interesting paper on an important and not fully clarified clinical problem,
i.e., the anterior knee pain following total knee arthroplasty. A newly method for
measuring patellar shift is introduced. Limitations of the study are the evaluation of
the parameters by a single observer as well as its retrospective nature. The use of 3
different TKA devices is another limitation.
1. Introduction
Good presentation of the background knowledge and the clinical question.
2. Material and methods
The number of patients included in the study is good if we consider OA as the main
problem but inadequate regarding the discrimination between primary and
secondary.
We thank the reviewer for this consideration. Unfortunately, it was not possible to
consider the form of osteoarthritis (primary or secondary) separately, as this would
have led to further reduction in group sizes and significantly reduced test power.
Use of three patient groups reduces significantly the statistical power.
We fully agree with the reviewer that larger groups would have significantly increased
the test strength, but the limitation e. g. to two patient groups would have been very
unbalanced in depicting the partly wide range of measured values, which is why we
decided to form a third "middle" group with predominantly normal values for better
comparability.

Normal values for the various indices regarding the sagittal alignment should be
provided.
We apologize for this unclarity. As suggested by the reviewer we now added the
normal value range for the each sagittal alignment parameter.
3. Results
No comments.
4. Discussion
Limitations are discussed, including the low number of complete f/up radiographs.
Please add the limitations suggested herein.
We thank the reviewer for this valuable observation of other limitations. We added
them to the limitation section.
Fig. 6. Please redraw the left red line.
We thank the reviewer for this interesting suggestion. Instead of exchanging this
image by another perfectly evaluable image we added following to the image caption:
“Additionally, the image shows an example of possible difficulties in
determination of patellar facet tangent, in this case due to bony superimposition
at the medial facet.”
5. References
There are many errors, please check via Pubmed. In some references the DOI is
included, in others not. Please follow the instructions to the authors.
We would like to apologize for formatting errors in the references. The references
now have been corrected according to the instructions to the authors

Reviewer 4 Report

The present article reveals great clinical relevance with a valid methodology.
However, authors could improve the introduction and objectives. Overall, it is the weakest component of the article.
The aforementioned method by Gomes has no bibliographic reference. Was this method developed by the authors? It is not clear!
Regarding the tables: in table 1 the SD of the mean values is not shown. None of the tables show the number of patients in each group or the times. Authors are invited to improve the table legends.
Finally, the discussion can be more elaborate and the conclusions more specific.

Author Response

Dear reviewer,

The present article reveals great clinical relevance with a valid methodology.
However, authors could improve the introduction and objectives. Overall, it is the
weakest component of the article.
We thank the reviewer for his valued opinion. We have significantly revised the
relevant sections.
The aforementioned method by Gomes has no bibliographic reference. Was this
method developed by the authors? It is not clear!
We kindly apologize for any misunderstanding. Accordingly, we added a new citation
of the paper by Gomes in line 163.
Regarding the tables: in table 1 the SD of the mean values is not shown. None of the
tables show the number of patients in each group or the times. Authors are invited to
improve the table legends.
We thank the reviewer for these considerations. Due to the missing normal
distribution, it is not permissible to specify the mean and standard deviation.
Therefore, the median and range must be specified. We clarified this point in line
248f.
Finally, the discussion can be more elaborate and the conclusions more specific.
We thank the reviewer very much for this suggestion. We have made multiple
revisions to this sections and hope that it now meets the high standards of the
esteemed reviewer.

Round 2

Reviewer 2 Report

supports earlier views. there are a lot of articles on related topics. The article does not add anything new.

Reviewer 3 Report

Thank you for addressing the comments.